# Reduced North Pacific Deep Water formation across the Northern Hemisphere Glaciation

Friso de Graaf [1,2] ✉, Heather L. Ford [1], Natalie Burls [3], Rachel Brown[4], Chris Brierley [2], Gavin L. Foster[5] & David Thornalley [2]

Today, deep waters do not form in the northern high latitudes of the Pacific Ocean, but this may not have been the case during the Pliocene. Evidence suggests there was a Pacific Meridional Overturning Circulation during the warm Late Pliocene, similar to the modern Atlantic Ocean with a weak halocline in the subpolar North Pacific resulting in North Pacific Deep Water (NPDW) formation. However, much of this evidence comes from proxies that can be biased by changes in local productivity. We present a coupled Mg/Ca-$\delta^{18}$O record from the North Pacific which shows two distinct water masses in the Pliocene North Pacific Ocean, with NPDW colder and fresher than the underlying deeper water. Here, we show a decline in NPDW formation during glacials from 2.7 million years ago, which we suggest demonstrates the strong sensitivity of ocean gateways to sea level and ice volume change in shaping deep water circulation, and thus the climate system.

Deep water circulation is an important component of the Earth's climate, moving heat, salt, and nutrients around the globe. Today, the formation of deep waters occurs in the high latitude North Atlantic and Southern Oceans[1]; there is no deep water formation in the North Pacific due to a strong upper ocean salinity gradient, termed a halocline[2,3]. This results in a homogenous deep Pacific Ocean with deep waters supplied from the Southern Ocean which travel northwards, lose density, upwell, and return to the Southern Ocean as surface and intermediate waters[1].

Though there is no North Pacific Deep Water (NPDW) formation at present, there is evidence for active deep water formation in the geologic past. Studies suggest there may have been NPDW formation in the Paleogene[4], the recent deglacial[5] and the mid-Pliocene Warm Period[6,7] (mPWP, 3.26–3.03 million years ago, Ma). Here we examine the history of NPDW formation over the intensification of Northern Hemisphere Glaciation (iNHG, c. 2.7 Ma) to investigate the sensitivity of deep water circulation to climate change as the Earth's climate transitioned from a warm climate to a cold one dominated by ice sheet growth and sea level change. Whilst there is evidence for NPDW formation in the Late Pliocene, this is absent in the Early Pleistocene[7]

suggesting that the iNHG played a role in suppressing this deep water formation. Deep water formation in the North Pacific would have markedly changed the global carbon cycle, increasing global export productivity by 20%[8], as well as altering patterns of ocean circulation in the Pacific[9] and the Atlantic[10], so it is important to understand how past deep water formation has changed in past warm climates. Active deep water formation in the North Pacific would require the removal of the halocline in the subpolar North Pacific, and studies have suggested a weak meridional (north-south) sea surface temperature (SST) gradient[7,11], changing monsoon patterns[12], and sea level induced changes in ocean gateways[13,14] as potential causes.

The evidence for NPDW formation in the Late Pliocene is contested and relies heavily on proxy records that can be biased by changes in local productivity. Many modelling studies of the Late Pliocene fail to generate an active overturning circulation in the North Pacific[15]; though this may be due to limited model run time[6], poor reconstructions of Late Pliocene temperature gradients[7], or an inability of the models to accurately simulate changing Pliocene palaeogeography[14]. Modelling studies that alter cloud physics to simulate a weak meridional Pacific SST gradient, as proxy data suggest

[1]School of Geography, Queen Mary University of London, London, UK. [2]Department of Geography, University College London, London, UK. [3]Atmospheric, Oceanic and Earth Sciences Department, George Mason University, Fairfax, VA, USA. [4]Centre Européen de Recherche et d'enseignement des géosciences de l'environnement (CEREGE), Aix-en-Provence, France. [5]School of Ocean and Earth Science, University of Southampton, National Oceanography Centre Southampton, Southampton, UK. ✉e-mail: f.m.degraaf@qmul.ac.uk

was the case during the Pliocene[11], generate active NPDW formation[7]. This water mass structure is supported by carbon isotopes across the Late Pliocene Pacific Ocean which point to NPDW reaching depths of around 3000 m in the North Pacific[6], though recent work from the subpolar North Pacific questions this[16]. Critically, these records come from opposite sides of the Pacific Ocean and NPDW formation is expected to occur in the Northwest Pacific Ocean during the Pliocene and is unlikely to show a similar water mass structure in the Northeast Pacific[9,17]. Carbon isotopes are also readily influenced by changes in local productivity[18] and so it is hard to determine which, if any, of these signals are driven by water mass changes. The presence of NPDW in the Pliocene is also supported by opal and carbonate mass accumulation rates[19,20] and redox-sensitive trace metal records[7] in the subpolar North Pacific, which both suggest an unstratified and well-ventilated water column indicative of active deep water formation as well as upwelling of deep waters in the Late Pliocene[7].

This study uses oxygen isotope ratios and minor element concentrations in benthic foraminifera to determine intermediate and deep water mass properties in the North Pacific during the Late Pliocene and Early Pleistocene. Foraminiferal $\delta^{18}O$ values are a function of temperature and the $\delta^{18}O$ of seawater ($\delta^{18}O_{sw}$), itself a function of salinity, with colder and/or more saline water masses having more positive (heavier) $\delta^{18}O$ values[18]. Therefore, $\delta^{18}O$ in benthic foraminifera ($\delta^{18}O_{benthic}$) can be used as a proxy for water mass density, with denser waters having more positive $\delta^{18}O$ values. However, it is possible for $\delta^{18}O_{sw}$ to be decoupled from salinity, often due to sea ice formation, such as is observed in the modern Southern Ocean[21]. Here we use the magnesium to calcium (Mg/Ca) ratios in benthic foraminifera as an independent proxy for bottom-water temperature (BWT) which is coupled with $\delta^{18}O_{benthic}$ measurements to deconvolve the temperature and salinity ($\delta^{18}O_{sw}$) contributions[22]. Additionally, $\delta^{18}O$, $\delta^{18}O_{sw}$ and Mg/Ca-derived temperatures are conservative tracers of water masses and so can be used to determine past water mass distributions; this contrasts with seawater stable carbon isotope ratios, which behave non-conservatively due to organic carbon remineralisation processes[23].

## Results and discussion
### Late Pliocene North Pacific Deep Water formation

Oxygen isotopes across a depth transect in the Northwest Pacific indicate there were two distinct water masses in the Late Pliocene North Pacific. We generated a $\delta^{18}O_{benthic}$ record at Ocean Drilling Program (ODP) Site 1209 (32°39.10' N, 158°30.36' E, 2387 m depth), from the Shatsky Rise in the Northwest Pacific and compared it to the nearby, but deeper, ODP site 1208 (3346 m depth[24]) (Fig. 1). During the Late Pliocene (3.3–2.7 Ma), the $\delta^{18}O$ values at the two sites show a mean difference of 0.25‰, but this is regularly in excess of 0.4‰ (Fig. 2). This is a significant difference in the $\delta^{18}O$ values ($p < 10^{31}$, $N = 601$) far in excess of the measurement uncertainty of ±0.05‰ and does not vary significantly between glacials and interglacials over the Late Pliocene ($p = 0.8110$, $N = 174$). Counter-intuitively, we report heavier $\delta^{18}O$ values at the shallower Site 1209 compared to the deeper Site 1208, suggesting denser water at Site 1209; as this is impossible, there must have been two distinct water masses, where one (or both) had a decoupled $\delta^{18}O_{sw}$-salinity relationship bathing the two different sites. We suggest these two water masses were a NPDW, which bathed Site 1209, and a southern-sourced deep water, which likely had a decoupled $\delta^{18}O_{sw}$-salinity relationship, similar to today, and flowed over the deeper site 1208, akin to the modern North Atlantic where AABW underlies NADW despite lighter $\delta^{18}O$[25].

This arrangement of water masses seen in the Late Pliocene North Pacific is unlike that of the present Pacific Ocean. Across the modern day North Pacific, core-top benthic foraminiferal $\delta^{18}O$ values are relatively homogenous compared to other ocean basins[26] and show a positive depth gradient with $\delta^{18}O_{benthic}$ values increasing by 0.10 ± 0.03‰ per km ($r = 0.30$, $p < 0.001$)[27], as opposed to the

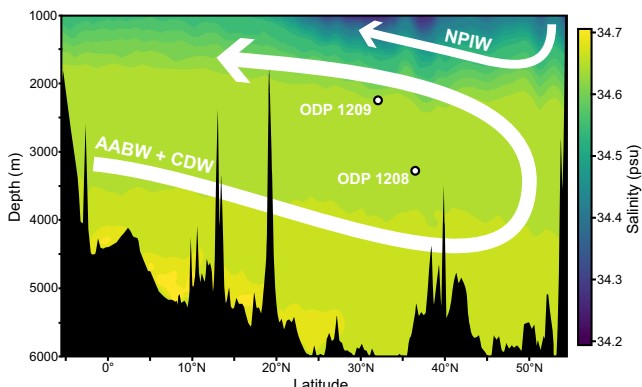

**Fig. 1 | Location of ODP Sites 1208 and 1209 and major water masses in the modern Pacific Ocean.** Salinity of the modern Pacific Ocean below 1000 m depth along WOCE Transect P13 running at 165°E from 4°S to 54°N[57]. Locations of the Ocean Drilling Program (ODP) Sites 1208 and 1209 are shown as circles. Major water mass trajectories in the North Pacific are shown with white arrows. AABW Antarctic Bottom Water, CDW Circumpolar Deep Water, NPIW North Pacific Intermediate Water. Bathymetry derived from eWOCE online archive[58].

decreasing $\delta^{18}O_{benthic}$ values with depth seen at the Shatsky Rise in the Late Pliocene. Modern $\delta^{18}O_{sw}$ values in the deep North Pacific (between 1000 and 4500 m depth) lie within −0.18‰ to −0.04‰ (2σ), a range of 0.14‰, and also show a positive depth gradient with $\delta^{18}O_{sw}$ values increasing by 0.021 ± 0.005‰ per km ($r = 0.46$, $p = 0.0001$)[28]. On the Shatsky Rise, core-top foraminiferal $\delta^{18}O$ values from Site 1209 (this study) and 1208[29] show no significant differences during the last 25 ka ($p = 0.678$, $N = 22$, Supplementary Fig. 3). Modern measurements of salinity and temperature on the Shatsky Rise show similar values at the two sites[30] (Fig. 3), with fresher (by 0.004 psu) and warmer (by 0.28 °C) waters at Site 1209. From this, we conclude the Shatsky Rise was not bathed by a persistent NPDW in the recent past.

Our coupled Mg/Ca and $\delta^{18}O_{benthic}$ values show that the NPDW was colder than the underlying southern-sourced waters during the Late Pliocene (Fig. 4). Our Mg/Ca-derived bottom water temperature (BWT) values at Site 1209 are colder than BWT values at Site 1208[29] over the period 3.0–2.7 Ma, with a difference in the mean BWT of 0.45 °C. When the BWT is coupled with the $\delta^{18}O_{benthic}$ values the colder Pliocene waters at Site 1209 must, therefore, have been fresher than those at the deeper Site 1208 to ensure density balance–in agreement with modelling evidence showing that the NPDW was fresher than underlying southern-sourced waters[7] (Supplementary Fig. 4). The derived $\delta^{18}O_{sw}$ values at Site 1209 are significantly ($p < 10^{10}$, $N = 107$) heavier than those at Site 1208 by 0.24‰ in the Late Pliocene (Fig. 4). Given the heavier $\delta^{18}O_{sw}$ at Site 1209 we therefore infer that the NPDW and southern-sourced deep water formation regions had different salinity-$\delta^{18}O_{sw}$ relationships, with one or both sites showing a decoupling of $\delta^{18}O_{sw}$ from salinity, likely due to sea-ice formation[31] (Fig. 3). Our results are constrained by the uncertainty in the BWT estimates, calculated by replicate analysis to be ± 0.8 °C. This uncertainty is greater than the difference in Mg/Ca-derived BWT between 1209 and 1208, with only a few instances (e.g., MIS G9 and G8) during the Late Pliocene where the difference in the inferred temperatures is greater than the uncertainty (Fig. 4).

It has been suggested that NADW may have been advected into the North Pacific during the Late Pliocene[32], which could thus be an explanation for the two distinct water masses revealed by the $\delta^{18}O_{benthic}$ records. Reconstructed NADW temperature was consistently warmer during the Late Pliocene than deep Pacific waters sourced from the Southern Ocean[29]. However, our BWT record shows that the temperatures measured at Site 1209 were consistently colder, not warmer, than the temperatures at Site 1208 (Fig. 5). We therefore

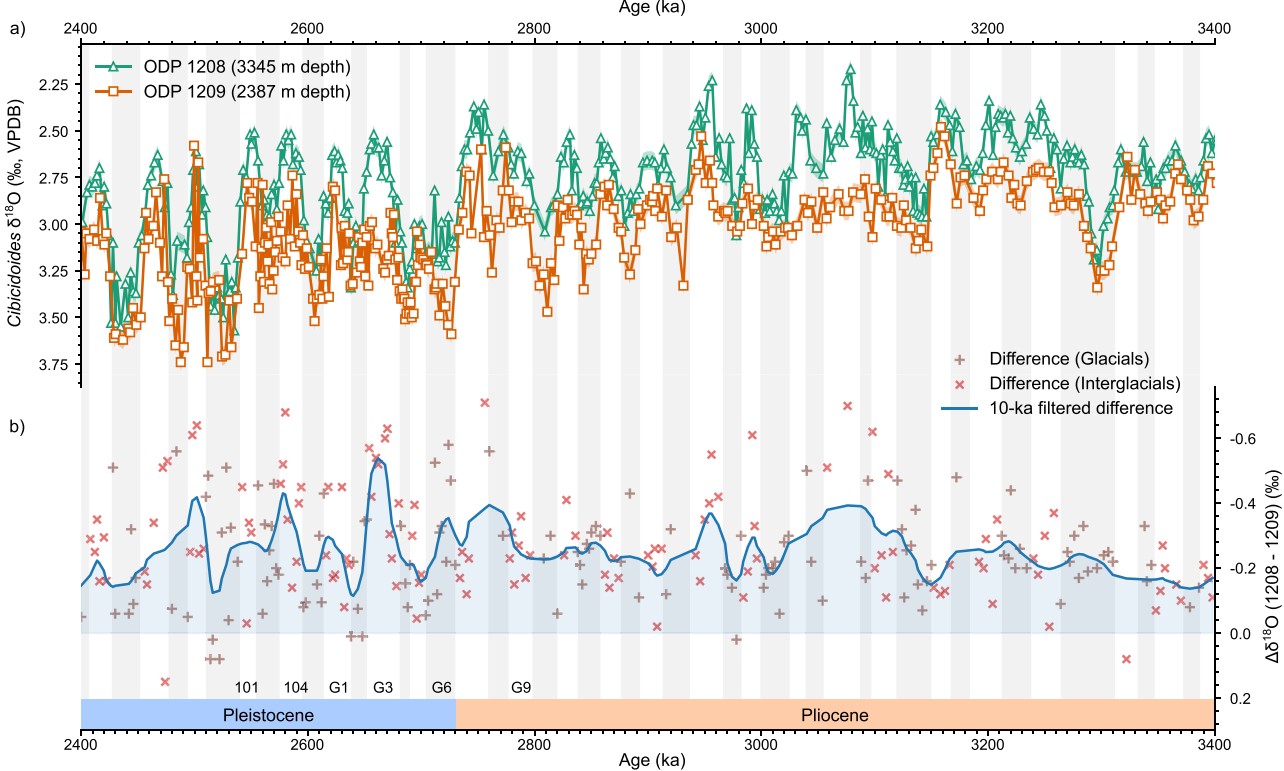

**Fig. 2 | Benthic oxygen isotope records of sites 1208 and 1209. a** Benthic isotope $\delta^{18}O$ records from ODP Site 1209 (2387 m depth, orange squares), and 1208[24] (3345 m depth, green triangles) over the iNHG (intensification of Northern Hemisphere Glaciation, 3.6–2.4 Ma). Error envelopes show ±0.05‰ uncertainty. **b** The difference in mean benthic oxygen isotope values between ODP Site 1209 and 1208 taken at 3-ka intervals for glacials (brown vertical crosses) and interglacials (red diagonal crosses) run through a 10-ka second order Butterworth low-pass filter (blue). A positive difference indicates more positive $\delta^{18}O$ values at Site 1209. Greater differences in $\delta^{18}O$ are thought to be indicative of a stronger North Pacific Deep Water (NPDW) influence at the shallower Site 1209. Glacial marine isotope stages according to the LR04[59] age model are shaded in grey, and certain key intervals are annotated. Source data are provided as a Source Data file.

interpret the more positive $\delta^{18}O$ values and colder BWT at Site 1209 compared to Site 1208 as a signal of NPDW formation and export to c. 2500 m depth North Pacific in the Late Pliocene, not incursion of NADW into the Pliocene Pacific Ocean.

**Early Pleistocene changes in deep water masses**

After iNHG (c. 2.7 Ma), the $\delta^{18}O_{benthic}$ records on the Shatsky Rise continue to be offset during interglacials (Fig. 2) and converge during glacials (Fig. 4) indicating that NPDW formation may have been suppressed during post-iNHG glacials. From MIS G6 onwards, the mean $\delta^{18}O_{benthic}$ difference decreased to 0.15‰ during Early Pleistocene glacials from 0.25‰ in the Late Pliocene (Figs. 2 and 6). During interglacials, however, the mean difference in $\delta^{18}O_{benthic}$ values actually increases to 0.32‰ during the Early Pleistocene, an increase of 0.07‰ from the Late Pliocene (Figs. 2 and 6). The difference in mean $\delta^{18}O_{benthic}$ shows no significant difference ( < 0.01‰, $p = 0.811$, $N = 174$) between glacials and interglacials in the Late Pliocene, but a significant difference (0.17‰, $p = 0.002$, $N = 107$) between glacials and interglacials during the Early Pleistocene. This suggests that there is some change over the iNHG which results in glacial-interglacial cycles modulating the formation and export of NPDW to intermediate depths of the North Pacific in the early Pleistocene. The difference in $\delta^{18}O$ values after the iNHG is strongly correlated with sea level estimates (Supplementary Fig. 5). Falling sea levels during glacials would close, or at least markedly reduce water mass transport through a number of oceanic gateways, which could have been a control on NPDW formation.

The Mg/Ca record for the Early Pleistocene also suggests that there is a convergence in water mass properties in the North Pacific during glacials but not interglacials. Over the iNHG, the mean derived

BWT at Sites 1209 and 1208 increased by 1.0 ± 0.8 °C and 1.1 ± 0.8 °C respectively (Fig. 4). This is likely due to a greater influx of warm NADW into the Southern Ocean and a greater stratification of Southern Ocean surface waters following the iNHG which would warm the southern sourced waters flowing into the Pacific[29,33]. Due to the reduction in NPDW formation after the iNHG, both sites would have experienced a greater share of Southern Ocean sourced waters which would have increased the BWT at both sites. While at Site 1209, there was effectively no difference in mean BWT between Early Pleistocene glacials and interglacials (0.7 °C and 0.6 °C respectively), Site 1208 saw a 0.5 °C difference develop between mean glacial and interglacial BWT (1.0 ± 0.8 °C and 1.4 ± 0.8 °C respectively). There was a slight decrease in the mean BWT gradient between Site 1209 and 1208 during Early Pleistocene glacials (0.3 °C) and an increase in the mean BWT gradient during interglacials (0.8 °C). This supports the idea of a convergence of water mass properties during glacials and a divergence during interglacials as seen in the $\delta^{18}O_{benthic}$ record, and lends credence to the idea that glacial-interglacial changes in the Early Pleistocene exerted a control on water mass formation in the Pacific that they did not during the Late Pliocene. These results are caveated by the high uncertainties in BWT estimates (± 0.8 °C), however, there are stretches of MIS 101, 103, G1 and G3 where the BWT estimates at 1208 are warmer than 1209 outside of uncertainties (Fig. 4) which is not seen during glacials, supporting the idea that there is some glacial-interglacial control on water mass properties in the deep North Pacific during the Pleistocene.

Our study shows an arrangement of $\delta^{18}O_{benthic}$ and BWT-reconstructions in the Late Pliocene that is explained by two distinct water masses in the intermediate and deep Northwest Pacific Ocean. This deep water mass is not present in the modern day or the Late

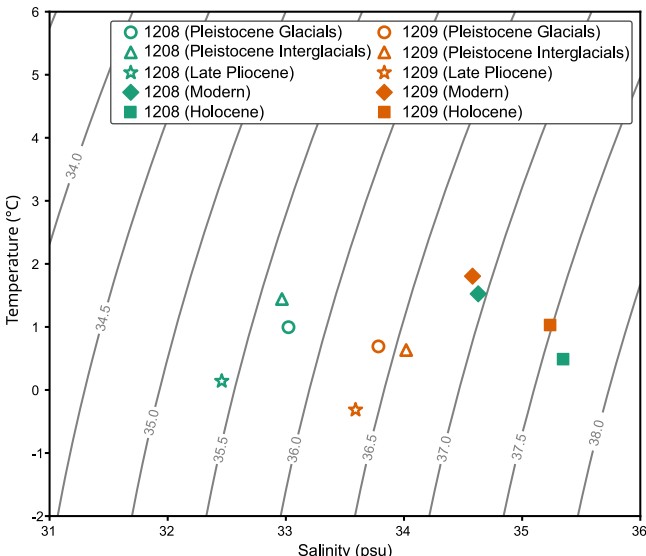

**Fig. 3 | Derived densities of past water masses requires the decoupling of past d$^{18}$O$_{sw}$-salinity.** Temperature and salinity of past water masses from ODP site 1208 (green, 3346 m depth) and 1209 (orange, 2387 m depth) for Late Pliocene (stars), Early Pleistocene glacials (circles) and Early Pleistocene interglacials (triangles). The temperature is the mean of the bottom water temperature (BWT) estimates for each site over all intervals for the Late Pliocene (2.7–3.0 Ma) and the Early Pleistocene (2.4–2.7 Ma). The mean of all the δ$^{18}$O$_{sw}$ estimates from the sites are used to determine the salinity estimates (see Methods). Isopycnals of σ$_2$ are calculated for 2300 m depth in the North Pacific using the Gibbs-Seawater Oceanographic Toolbox[60] and shown in grey and labelled. The densities of waters at 1209 are greater than those at (deeper) site 1208 for the Late Pliocene and Early Pleistocene (hollow symbols) which suggests that the derived δ$^{18}$O$_{sw}$ values for 1208 and/or 1209 cannot be linearly converted to past salinities. Modern values of temperature and salinity (diamonds) from Argo float data[30], and calculated temperatures and salinities from core-top mean δ$^{18}$O$_{benthic}$ and Mg/Ca values from 1209 and 1208[29] for the Holocene (0–12 ka, squares) are shown highlighting that modern and Holocene δ$^{18}$O$_{sw}$ measurements do appear correlated to salinity (filled symbols). Source data are provided as a Source Data file.

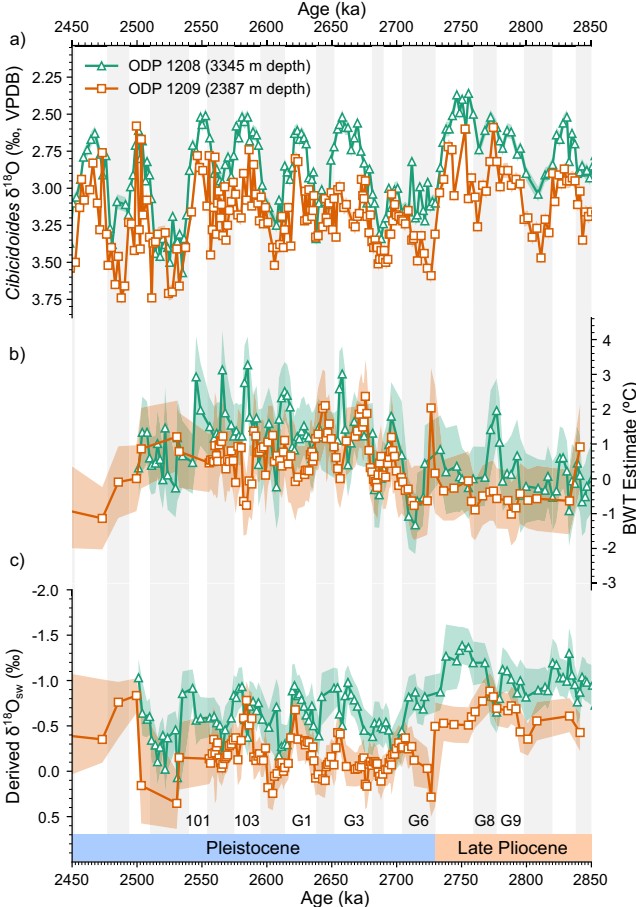

**Fig. 4 | Derived temperatures and seawater δ$^{18}$O records of 1208 and 1209.** **a** Benthic isotope δ$^{18}$O records from ODP Site 1209 (2387 m depth, orange squares), and 1208[24] (3345 m depth, green triangles) over the period 2.9–2.4 Ma. **b** Bottom-water temperatures derived from Mg/Ca measurements from 1209 (orange squares) and 1208[29] (green triangles). **c** Seawater δ$^{18}$O records from coupled Mg/Ca and benthic δ$^{18}$O measurements for Site 1209 (orange squares) and 1208[29] (green triangles). Orange and green shading indicates 95% confidence interval. Glacial marine isotope stages according to the LR04[59] age model are highlighted and annotated. Source data are provided as a Source Data file.

Pleistocene[34] which means the strong subpolar North Pacific halocline seen in the modern Pacific likely formed during the Early Pleistocene. The glacial-interglacial variability in the difference in δ$^{18}$O$_{benthic}$ and BWT after the iNHG (Fig. 6), and the correlation between the difference in δ$^{18}$O$_{benthic}$ and Late Pleistocene sea levels (Supplementary Fig. 5) suggests this process is modulated by iNHG ice sheet expansion.

We suggest falling sea levels in Early Pleistocene glacials would restrict water mass transport through the Indonesian Gateway and act as a control on the formation of NPDW. Closing the Indonesian Gateway would halt NPDW formation by increasing the precipitation minus evaporation in the North Pacific[14]. During the Pliocene, warm, saline waters flowed through the Indonesian Gateway originating from the Equatorial and Southern Pacific Ocean[35,36]. The closure of the Indonesian Gateway would redirect this warm water back to the Equatorial Pacific, warming it relative to the high latitudes[37], which would increase the meridional (north-south) SST gradient in the Pacific Ocean. This would increase the precipitation minus evaporation in the North Pacific[7,38], decrease surface salinity in the North Pacific, strengthen the subpolar halocline, and thus halt NPDW formation (Fig. 7). Alkenone SST records show a c. 1 °C stronger Pacific meridional SST gradient after the iNHG[39,40] (Supplementary Fig. 6). Stronger SST gradients during MIS G2, 100, and the first half of MIS 104 correspond to reduced δ$^{18}$O gradients, while interglacials with weaker SST gradients, such as MIS G3 and 99, see greater δ$^{18}$O gradients (Supplementary Fig. 6). Similarly, planktic oxygen isotope records from the subpolar North Pacific imply local salinity changes, proportional to the strength of the

subpolar halocline, in phase with global sea level changes after 2.7 Ma[41]. The Indonesian Gateway thus links changes in sea level to the North Pacific meridional SST gradient and subpolar surface salinity, and the 1208–1209 δ$^{18}$O gradient after the iNHG.

The water mass transport through the Indonesian Gateway, modulated by global sea levels changes, is unlikely to be the cause of other instances of NPDW formation in the geologic past. We suggest that although the halocline is eroded during the Pliocene, the resultant NPDW is still fresher than underlying southern-sourced waters. The minimal impact of BWT changes on the density of water masses at Sites 1208 and 1209 highlights the importance of salinity in determining the formation of NPDW (Fig. 3). In the Late Pleistocene, there are suggestions of deep water formation during the LGM[5], but in contrast to the Pliocene this was triggered by regional cooling and increased surface salinity in the North Pacific, in part aided by a closed Bering Strait due to lower sea levels[5,13,42]. That falling sea levels can halt NPDW formation over the iNHG, yet low sea level induces NPDW formation in the Late Pleistocene is likely due to different heat and freshwater balance over these timescales.

Our results show the presence of two distinct water masses in the Late Pliocene North Pacific which we interpret to be NDPW and a Southern Ocean-sourced water mass. During the Pleistocene the signal

of NPDW formation weakens during glacials but becomes stronger during interglacials, suggesting some modulation of deep water formation in the North Pacific by glacial-interglacial factors that is not present prior to the iNHG. We suggest that this could be due to falling sea levels reducing the water mass transport through the Indonesian Gateway after the iNHG which would have wide-reaching effects on the Pacific climate. After iNHG, there is a convergence in deep Pacific and Atlantic Ocean characteristics[29], suggesting greater homogeneity and connectivity in global deep water characteristics in the Pleistocene compared to the Pliocene. This may be driven by the cessation of NPDW formation allowing for a greater Antarctic Bottom Water extent, as well as stronger Antarctic Bottom Water export from the Southern Ocean following the growth of the Antarctic ice sheet[43]. This reorganisation in deep ocean circulation had cascading impacts on the carbon cycle and likely resulted in greater inorganic carbon storage influencing atmospheric temperatures and carbon cycling in the Pleistocene[12].

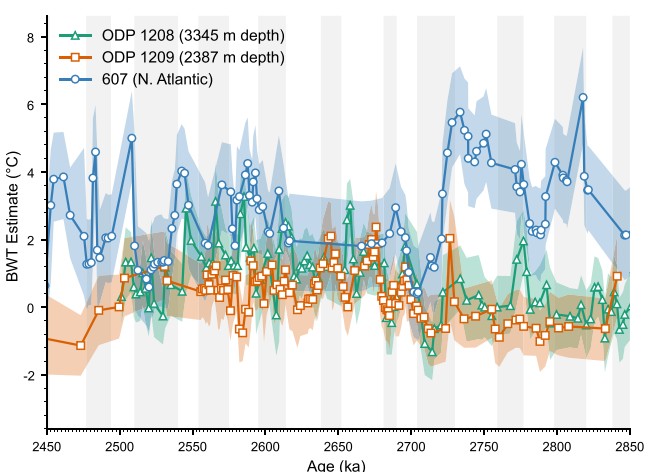

**Fig. 5 | Comparison of North Pacific and North Atlantic BWT records.** Bottom-water temperatures derived from Mg/Ca measurements using PSU Solver (see methods), from Site 1209 (orange squares), 1208[29] (green triangles), and 607[33] (blue circles). Orange, green, and blue shading indicates 95% confidence interval. Glacial marine isotope stages according to the LR04[59] age model are highlighted. Source data are provided as a Source Data file.

## Methods

### Sample preparation
Ocean Drilling Program (ODP) Leg 192 Site 1209 is located on the Shatsky Rise in the NW Pacific Ocean (32°39.11′N, 158°30.36′E) at 2387 m depth[44]. Samples were washed in deionised water and benthic foraminifera were picked from the >150μm size fraction for geochemical analysis.

### Stable Isotopes
Four hundred twenty-five samples of the benthic foraminifera *Cibicidoides wuellerstorfi* were used for stable isotope analysis. Approximately 5–7 specimens were analysed at c. 4 cm resolution along the core for both oxygen and carbon isotopes using a VG ISOGAS SIRE dual inlet isotope ratio mass spectrometer at the Godwin Laboratory for Palaeoclimate Research at the University of Cambridge. The long term analytical reproducibility for NBS-19 is ±0.06‰ and 0.05‰ (1 s.d.) for carbon and oxygen isotopes. All stable isotope data are reported relative to Vienna Pee Dee Belemnite (V-PDB) using the conventional notation. Calibration to reference standards allows for comparison of these records to stable isotope records from Site 1208[24]. Previous stable isotope analysis was also undertaken on the species *Uvigerina peregrina* for the two sites[6,29] at lower resolution and shows the same pattern of heavier oxygen isotopes at the shallower Site 1209 (Supplementary Fig. 8).

### Trace metal analysis
The oxygen isotope measurements were paired with Mg/Ca measurements from the same depth in the core. 135 samples of between 8 and 12 foraminifera of the species *Uvigerina peregrina* were picked for trace metal analysis. In some samples of greater abundance, replicate analysis was performed ($n = 9$). The samples were crushed between two glass slides. They then underwent a modified reductive and oxidative cleaning process described in ref. 45,46 respectively. Elemental analyses were performed on a Thermo-Fisher Scientific Element 2 single collector sector field ICPMS at University of Southampton following established methods[47]. For this study, analytical reproducibility for Mg/Ca was ±3% (2σ) based on repeat measurements of inhouse carbonate standards.

The samples were measured for Mg/Ca as well as Mn/Ca, Fe/Ca and Al/Ca to monitor for any contamination of the sample from remnant diagenetic coatings or detrital material. Any samples with values of Mn/Ca, Fe/Ca, or Al/Ca over 120 μmol/mol were rejected (Supplementary Fig. 1). There is no correlation or temporal relationship

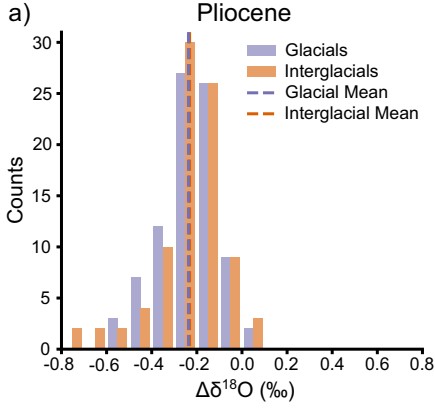

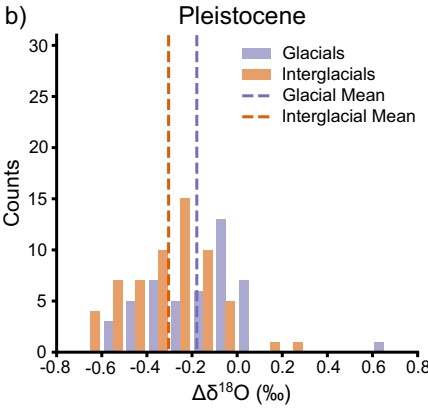

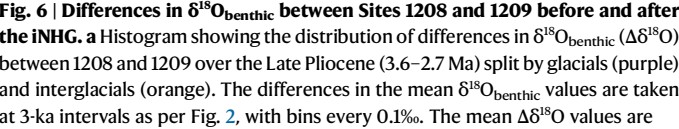

**Fig. 6 | Differences in δ18Obenthic between Sites 1208 and 1209 before and after the iNHG. a** Histogram showing the distribution of differences in δ18Obenthic (Δδ18O) between 1208 and 1209 over the Late Pliocene (3.6–2.7 Ma) split by glacials (purple) and interglacials (orange). The differences in the mean δ18Obenthic values are taken at 3-ka intervals as per Fig. 2, with bins every 0.1‰. The mean Δδ18O values are shown as dashed lines. The glacial mean value is −0.24‰ and the interglacial mean value is −0.23 ‰. **b** Histogram showing the distribution of Δδ18O for the Early Pleistocene (2.7–2.4 Ma) for glacials (purples) and interglacials (orange) showing the mean Δδ18O values as dashed lines. The glacial mean value is −0.18 ‰, and the interglacial mean value is −0.30‰. Source data are provided as a Source Data file.

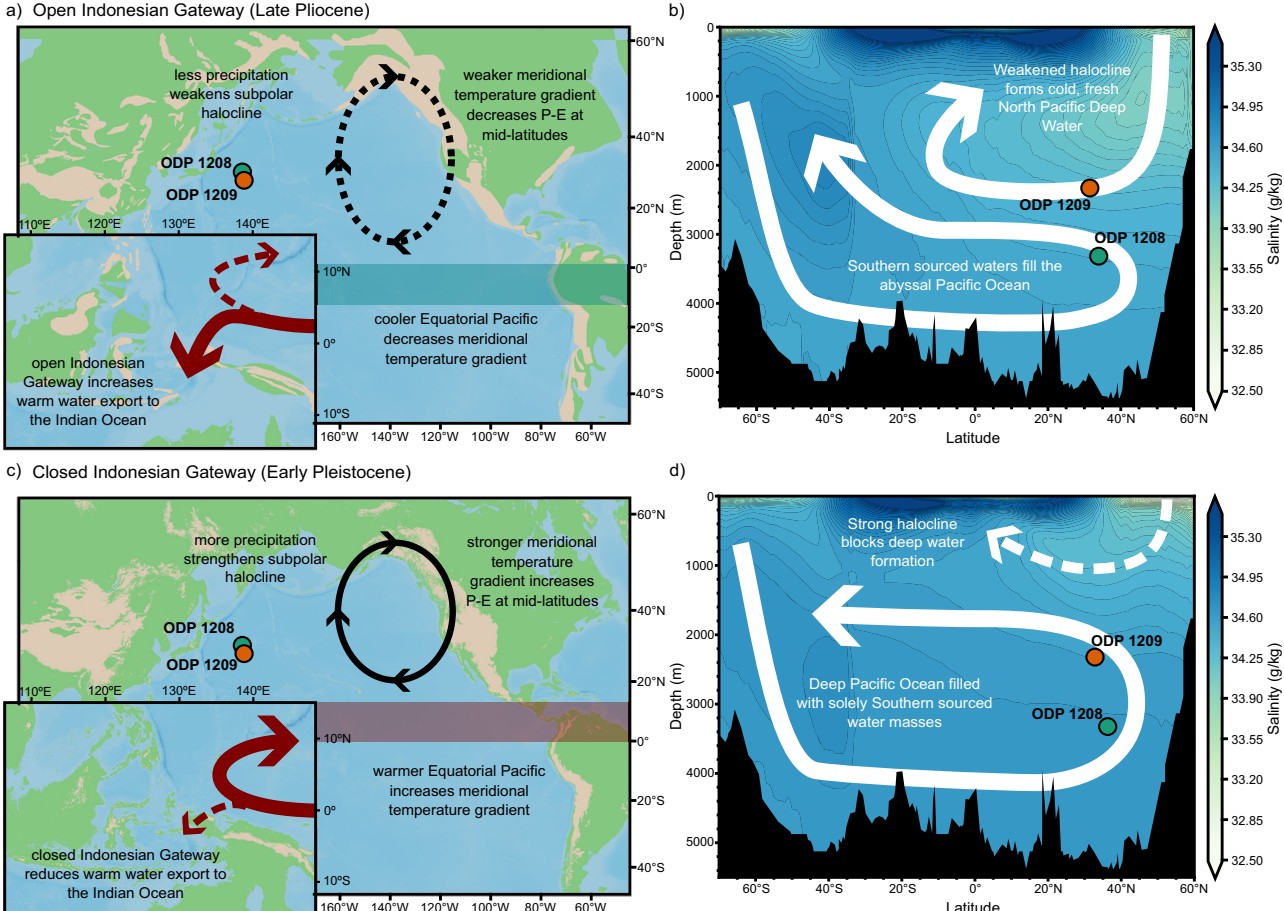

**Fig. 7 | Schematic showing how glacial sea level change could modulate Pacific Ocean circulation. a** Schematic diagram showing ocean circulation in the Pacific Ocean during the Late Pliocene with mid Pliocene sea levels highlighting how an open Indonesian Gateway could promote NPDW formation; a weaker meridional temperature gradient would decrease precipitation−evaporation (P-E) over the mid latitudes and weaken the subpolar halocline. **b** Ocean circulation in the deep Pacific Ocean from the Late Pliocene with the relative extent of NPDW and Antarctic Bottom Water shown. Shading shows modelled salinity of the Late Pliocene Pacific Ocean[7]. **c** Schematic diagram showing ocean circulation in the Pacific Ocean during glacials of the Early Pleistocene with sea levels 60 m lower than modern, highlighting how closing the Indonesian Gateway strengthens the North Pacific halocline. **d** Ocean circulation in the deep Pacific Ocean from the Early Pleistocene. Shading shows modelled salinity of the Pre-Industrial Pacific Ocean[7]. Palaeogeography based on Pliocene map of ref. 61. Pleistocene geography based on modern coastlines and topography[62] with sea levels 60 m below modern. Elevation above 1000 m is shown in brown. Location of ODP Sites 1209 (orange) and 1208 (blue) are shown as circles.

between Mn/Ca, Fe/Ca or Al/Ca and Mg/Ca (Supplementary Fig. 2). Replicate analysis ($n = 9$, separate populations of cleaned foraminifera) have an uncertainty in the Mg/Ca measurements of $\pm 0.085$ mmol/mol (1 s.d.).

PSUSolver[48] was used to calculate temperature and seawater $\delta^{18}O$. Using the Mg/Ca and $\delta^{18}O$ values, PSUSolver uses a Monte Carlo model to simultaneously solve paleotemperature equations and propagate uncertainty. The Mg/Ca-temperature equation used in the model was from ref. 22 for *Uvigerina* with a 10% adjustment to account for the reductive cleaning step. The temperature-$\delta^{18}O$ equation was from ref. 49 for *Cibicidoides wuellerstorfi*. The uncertainties used were $\pm 0.085$ for the Mg/Ca (the uncertainty for replicate analysis) and $\pm 0.05$ for $\delta^{18}O$.

## Age model
An orbitally tuned age model for Site 1209 was made with HMM-Stack[50]. We used the HMM-Stack age estimator to tune our record using a probabilistic Bayesian model to align the benthic oxygen isotope record to ProbStack, a probabilistic global benthic $\delta^{18}O$ stack[50]. This was run on Queen Mary's Apocrita HPC facility, supported by QMUL Research-IT[51]. Our sampling resolution is 1–4 kyr. The age model for the 1209 core-top samples was adapted from ref. 52.

## Change point analysis
The benthic $\delta^{18}O$ record was analysed using a Bayesian change point algorithm[53]. The algorithm was set up to look for the largest change point in the record, with a minimum spacing between points of 10 ka to avoid a single large glacial or interglacial peak being indicated as a change point.

## Salinity calculations
Salinity of past water masses is calculated using the $\delta^{18}O_{sw}$ estimates from PSU Solver (see above). The $\delta^{18}O_{sw}$ is first corrected for the effects of changing ice volume, $\delta^{18}O_{ivc}$, for the specific interval taken fromn ref. 54. The corrected $\delta^{18}O_{sw}$ is compared to the modern $\delta^{18}O_{sw}$ from the water depth and latitude in the North Pacific as in ref. 25, to get the difference in $\delta^{18}O_{sw}$. The difference in $\delta^{18}O_{sw}$ is converted to a difference in salinity using the North Pacific $\delta^{18}O_{sw}$-salinity relationship of $\Delta\delta^{18}O_{sw} = (0.44 \times \Delta\,Salinity)$ from ref. 25. The difference in salinity is then added to the modern salinity for the site to give an estimate of the past salinity of the site. This estimate of salinity is corrected for the change in salinity relationship from changing sea levels using the sea level curve from ref. 54. The difference in sea level, $\Delta RSL$, is divided by the modern average depth of the oceans, 3680 m, and then multiplied by the modern average salinity of the oceans, 34.7 psu, to give the

correction added to the salinity estimate to achieve the final salinity estimate.

## Statistical analysis

The significance of any similarities in datasets (in either the δ¹⁸O and Mg/Ca record for both Sites 1209 and 1208) was measured using a Welch's *T*-test. A standard *T*-test was not used as the δ¹⁸O or Mg/Ca records for both Sites 1209 and 1208 were not normally distributed. This was tested by a Kolmogorov–Smirnov test (comparing the distribution to a normal distribution) and a Shapiro–Wilk test, the results of these tests are provided Supplementary Table 1.

## Data availability

All data generated from sediment core ODP 1209 for this study have been made available in the Pangaea.de repository[55], where the files can be accessed under the following https://doi.org/10.1594/PANGAEA.969149. Source data are provided as a Source Data file. Source data are provided with this paper.

## Code availability

The CESM 1.2.2.1 code used to run the climate simulations is available from https://svn-ccsm-models.cgd.ucar.edu/cesm1/release_tags/cesm1_2_2_1. The code for generating the seawater density figure in Fig. 3 is available on GitHub at the following link https://doi.org/10.5281/zenodo.14871574[56].

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

## Acknowledgements

We thank D. Fairman, P. Goh and Y. Meng as well as J. Rolfe, J. Booth and the Godwin Laboratory for their laboratory assistance and the International Ocean Discovery Program core repository for providing samples. This work was supported by the London Natural Environmental Research Council (NERC) Doctoral Training Partnership (NE/S007229/1; F.dG.), NERC (NE/N015045/1; H.F. and NE/Y000781/1; CB). N.J.B. acknowledges funding support from National Science Foundation awards AGS-1844380 and OCE-2225829.

## Author contributions

F.dG. and H.L.F. designed the study. F.dG. and R.B. carried out the geochemical analysis. The CESM model was developed by N.B. The data was analysed by F.dG. with assistance from N.B., D.T., C.B., G.L.F., R.B. and H.L.F. F.dG., N.B., D.T., C.B., G.L.F., R.B. and H.L.F., contributed to the final manuscript.

## Competing interests

We declare that none of the authors have competing interests as defined by Nature Portfolio.
