## [Peer Review file · Nature Communications]

Reduced North Pacific Deep Water Formation across the Northern Hemisphere Glaciation

Corresponding Author: Mr Friso de Graaf

Version 0:

Reviewer comments:

Reviewer #1

(Remarks to the Author)

The paper "North Pacific Deep Water Formation controlled by Sea Level Changes" investigates ocean circulation during the Late Pliocene, focusing on the mechanisms and the controlling factors.

The paper provides new data on stable isotopes and Mg/Ca ratio on foraminiferal tests from Site 1209 (North Pacific) and compared it to the nearby site 1208, also from the North Pacific.

The comparison between the records of the two cores suggests that, in the comparison to the deeper Site 1208, colder and/or more saline water was present at the shallower Site 1209. Since the deepest site should be characterised by denser water, the authors determine that the only possible reason to cause this difference is the presence of two distinct water masses in the area, a North Pacific Deep Water, and a southern-sourced deep water, which must have had different temperatures and $\delta^{18}\text{O}$ values. Consequently, they analyze the record of the two cores using the differences between them as a proxy for the export of North Pacific Deep Water. They also discuss how this process might have happened during different time periods.

My research mainly focuses on shallow-water successions (although I have worked on stable isotopes and Mg/Ca ratio on foraminifera and therefore I know how variable these data can be). Therefore, I may not be accustomed to many paleoceanographic details. However, I find two major flaws in the paper that require a moderate revision of data presentation and a major revision of the text, namely insufficient presentation and analysis of the data and improper organization of the discussion.

The main focus of the paper is the difference between the $\delta^{18}\text{O}$ of the two cores intended as a proxy for the differences between the water masses. These differences are very subtle, especially in the way in which they are depicted into the graphs, and they are only qualitatively discussed.

In figure 2 the differences are associated with a shading that indicates the uncertainty. Although these shading are too faint to be properly appreciated, there is a clear superposition, especially in the estimated temperature of bottom waters. This type of superposition between the data clearly highlights that the differences are not so large. They are there, but they are not so clear. In my opinion this requires a little bit of statistical analyses to prove that the differences are statistically relevant (especially taking into account the variability of replicate analysis of each sample). On the other hand, from line 103 to 107 where the differences in $\delta^{18}\text{O}$ are discussed, the manuscript is quite generic:

"During MIS G4, G2, and the second half of MIS 104, there was a $<0.5^\circ\text{C}$ difference in BWT between the two sites, and a $<0.1\text{‰}$ difference in $\delta^{18}\text{O}$, compared to a $>0.4\text{‰}$ difference in $\delta^{18}\text{O}$ values in interglacials MIS G3, G1, and 103 (Figure 2)"

As the whole paper is based on this evidence, I believe that a more extensive discussion and a more detailed statistical tests of the relevance of these differences is required. In this regard I think that Extended Data Figure 4 is much more effective in conveying the differences between the water masses. Extended Data Figure 4 includes only some data (maybe averages values? In the caption an explanation is not provided), however, a similar figure with all the data and their scatter, coupled with a statistical analysis would be much more effective in conveying the results of the paper.

Indeed, following the brief discussion of the data from line 103 to line 107, the paper essentially becomes a review of literature data on various proxies related to ocean circulation during the Pliocene – Pleistocene interval. This discussion/review is much more extensive than the analysis of the data provided by the authors and relatively disconnected from the latter. Furthermore, this part is not properly organized and the narrative wanders between various subjects and elements (too many for a short paper) without a clear direction.

In order to improve its impact the paper needs to be better organized: 1) Better and more clear presentation of the data 2) More detailed analysis of the data and their significance, using statistical methods to test the significance of the observed differences taking into account all the uncertainties. 3) Shorter and more impactful discussion of the existing literature to support the evidence obtained from the analyzed dataset.

There are other minor flaws that should also be addressed (how many samples were analysed? In certain figures, like extended figure 4, it is unclear if average values or selected values are being displayed). I have provided some other small notes in the annotated version of the manuscript.

I hope that these suggestions might help the authors and the future impact of their paper to the general scientific community.

Best regards

Reviewer #2

(Remarks to the Author)

Review of the manuscript "North Pacific Deep Water Formation controlled by Sea Level Changes", by de Graaf et al.

The manuscript presents a very interesting benthic $\delta^{18}\text{O}$ record from the North Pacific (ODP 1209, at 2387 m depth) that, compared to the $\delta^{18}\text{O}$ record of a deeper site (ODP 1208, at 3345 m depth) from the same region, has been interpreted as a result of the presence of different water masses along the water column. The authors interpreted that ODP 1209 record indicates North Pacific deep water formation was active before the intensification of the Northern Hemisphere glaciation. They also linked the cessation of North Pacific deep water formation to the restriction in the water mass transport through the Indonesian Gateway, as suggested by ocean modelling studies.

The hypothesis proposed is interesting, however, I do not clearly see in the presented graphics a dataset that supports a clear cessation of North Pacific deep water formation after the intensification of the Northern Hemisphere glaciation. Figure 1 shows the difference benthic $\delta^{18}\text{O}$ difference between the two sites as a proxy for the strength of North Pacific deep water formation but there is no marked reduction. The discussion about BWT is not easy to follow in figure 2, a gradient would be better to show if there is a difference between both sites that changed at the intensification of the Northern Hemisphere glaciation. Same thing for $\delta^{18}\text{O}_{\text{w}}$. Figure 2d is intriguing, it seems that instead of a cessation in North Pacific deep water formation, this process was enhanced during interglacials after 2.7 Ma. The support for the presence/absence of North Pacific deep water formation must be clearly explained indicating the differences between the Southern Ocean deep water and the NPDW.

In addition, the text tries to justify the hypothesis from the very beginning. I think it would be better to present first the modern conditions and water masses characteristics; then indicate how the interval before the intensification of the Northern Hemisphere glaciation shows different conditions. The discussion about possible drivers for this change is also repeated in two different sections. I suggest reading again all the manuscript and try to arrange the story and the data in a more clear way.

Minor comments

Line 57: I believe the authors wanted to report that "A positive difference indicates more positive $\delta^{18}\text{O}$ values at Site 1209" (not in 1208)

Line 63: specify benthic foraminifers

Version 1:

Reviewer comments:

Reviewer #1

(Remarks to the Author)

The revised version of "Reduced North Pacific Deep Water Formation across the Northern Hemisphere Glaciation" significantly improved the shortcomings of the original version. The data are now better discussed, associated with a statistical analysis, and supported by clear figures, making the message conveyed by the manuscript much clearer and more impactful.

I believe that the paper is now suitable for publication. I have noted some small typos here and there in the annotated version of the manuscript.

I hope that this small correction can help the authors, and I hope to see the manuscript published soon.

Best regards

(Remarks on code availability)

Reviewer #3

(Remarks to the Author)

Response:

In regard to the North Pacific Deep Water formation in the Late Pliocene, there is some controversial on the proxy and model climate. In this work, Dr. Graaf et al. present a coupled Mg/Ca- $\delta^{18}\text{O}$ record from the North Pacific which shows two distinct water masses in the Pliocene North Pacific Ocean, with NPDW colder and fresher than the underlying deeper water. Using two cores from different water depth, they further prove the NPDW formation decline during glacials from 2.7 million years ago, which highlight the impact of sea level via ocean gateways in shaping deep water circulation in the climate system. After the new version and response letter, I think the authors have gave a good reply and made the modifications in the manuscript. I give some comments on this work.

Firstly, in the Abstract part: I suggest that the author should add some highlights in the last sentence in this part. In this version, I cannot get your key points in your study.

Secondly, I suggest that you should add a figure 1 about the location in the North Pacific Ocean. In this case, you can present the deep-water circulation in the North Pacific Ocean.

After some improvement in this part, I suggest this manuscript can be accepted after the minor revision.

(Remarks on code availability)

Note: Reviewer comments are in green text, author replies are in black text, and manuscript edits are in black text with track changes. The annotated line numbers refer to those lines in the final document.

We would like to thank the two reviewers for their helpful feedback, and we have changed the paper accordingly. The two reviewers commented that the central hypothesis of the paper, the formation of North Pacific Deep Water and its cessation over the iNHG, was not presented clearly enough either graphically or statistically, we have addressed these points in this new draft and believe that the paper is better for it. It was also noted that the structure of the paper could be made clearer, with a greater focus on the results, which we have done as well as adjusting the paper to fit the Nature Communications structure. In light of these changes, we have also changed the title of the paper, to “Reduced North Pacific Deep Water Formation across the Northern Hemisphere Glaciation”, to put the results at the centre of this paper rather than hypotheses for what may have caused them. We believe that this paper is now a more focused, data-led manuscript and would like to thank the reviewers for their helpful comments in getting the paper to this stage.

Individual points raised by the reviewers are addressed below.

Reviewer #1 (Remarks to the Author):

The paper “North Pacific Deep Water Formation controlled by Sea Level Changes” investigates ocean circulation during the Late Pliocene, focusing on the mechanisms and the controlling factors.

The paper provides new data on stable isotopes and Mg/Ca ratio on foraminiferal tests from Site 1209 (North Pacific) and compared it to the nearby site 1208, also from the North Pacific.

The comparison between the records of the two cores suggests that, in the comparison to the deeper Site 1208, colder and/or more saline water was present at the shallower Site 1209. Since the deepest site should be characterised by denser water, the authors determine that the only possible reason to cause this difference is the presence of two distinct water masses in the area, a North Pacific Deep Water, and a southern-sourced deep water, which must have had different temperatures and $\delta^{18}\text{O}$ values.

Consequently, they analyze the record of the two cores using the differences between them as a proxy for the export of North Pacific Deep Water. They also discuss how this process might have happened during different time periods.

We would like to thank the reviewer for their comments. We have edited the paper to

present the data more clearly, better highlight the significance of the results, and have restructured the discussion to better reflect this.

My research mainly focuses on shallow-water successions (although I have worked on stable isotopes and Mg/Ca ratio on foraminifera and therefore I know how variable these data can be). Therefore, I may not be accustomed to many paleoceanographic details. However, I find two major flaws in the paper that require a moderate revision of data presentation and a major revision of the text, namely insufficient presentation and analysis of the data and improper organization of the discussion.

The main focus of the paper is the difference between the $\delta^{18}\text{O}$ of the two cores intended as a proxy for the differences between the water masses. These differences are very subtle, especially in the way in which they are depicted into the graphs, and they are only qualitatively discussed.

This is a good point, and we have addressed this in three principal ways:

1. We have performed two-tailed Welch's T-tests on the $\delta^{18}\text{O}$ from 1208 and 1209, for the period before and after the iNHG to show quantitatively that the mean values are different before and after the iNHG. Furthermore, the differences between the two records before and after the iNHG are discussed more qualitatively, and the qualitative way in which the oxygen isotope records differed was removed.

During the Late Pliocene (3.3 – 2.7 Ma), the $\delta^{18}\text{O}$ values at the two sites show a mean difference of 0.25‰, but this is regularly in excess of 0.4 ‰ (Figure 1). This is a significant difference in the $\delta^{18}\text{O}$ values ($p < 10^{-31}$, $N = 601$) far in excess of the measurement uncertainty of ± 0.05 ‰ and does not vary significantly between glacials and interglacials over the Late Pliocene ($p = 0.8110$, $N = 174$). (Lines 91 – 95)

After iNHG (c. 2.7 Ma), the $\delta^{18}\text{O}_{\text{benthic}}$ records on the Shatsky Rise continue to be offset during interglacials (Figure 1) and converge during glacials (Figure 3) indicating that NPDW formation may be suppressed during post-iNHG glacials. From MIS G6 onwards, the mean $\delta^{18}\text{O}_{\text{benthic}}$ difference decreased to 0.15 ‰ during Early Pleistocene glacials from 0.25 ‰ in the Late Pliocene (Figures 1, 5). During interglacials, however, the mean difference in $\delta^{18}\text{O}_{\text{benthic}}$ values actually increases to 0.32 ‰ during the Early Pleistocene, an increase in 0.07 ‰ from the Late Pliocene (Figures 1, 5). The difference in mean $\delta^{18}\text{O}_{\text{benthic}}$ shows no significant difference (< 0.01 ‰, $p = 0.811$) between glacials and interglacials in the Late Pliocene, but a significant difference (0.17 ‰, $p = 0.002$) between glacials and interglacials in the Early Pleistocene. (Lines 147 – 155)

2. We have included a histogram of the distribution of $\delta^{18}\text{O}$ and BWT from 1208 and 1209 during glacials and interglacials before and after the iNHG in Figure 5 which shows the above results in a more intuitive form for the reader. As well as this, we

have removed the lines joining the binned mean difference in $\delta^{18}\text{O}$ values in Figure 1 (in grey) so that the 10-ka filtered difference is clearer which better highlights the glacial-interglacial disparity in $\Delta\delta^{18}\text{O}$ post-iNHG.

3. We have included a greater qualitative discussion of the modern arrangements in the deep North Pacific Ocean to make clear that the difference in oxygen isotopes in the Late Pliocene is not akin to modern day ocean dynamics

This arrangement of water masses seen in the Late Pliocene North Pacific is unlike that of the present Pacific Ocean. Across the modern day North Pacific, core-top benthic foraminiferal $\delta^{18}\text{O}$ values are relatively homogenous compared to other ocean basins²⁷ and show a positive depth gradient with $\delta^{18}\text{O}_{\text{benthic}}$ values increasing by 0.10 ± 0.03 ‰ per km ($r = 0.30$, $p < 0.001$)²⁸, as opposed to the decreasing $\delta^{18}\text{O}_{\text{benthic}}$ values with depth seen at the Shatsky Rise in the Late Pliocene. Modern $\delta^{18}\text{O}_{\text{sw}}$ values in the deep North Pacific (between 1000 – 4500 m depth) lie within -0.18 ‰ to -0.04 ‰ (2σ), a range of 0.14 ‰, and also show a positive depth gradient with $\delta^{18}\text{O}_{\text{sw}}$ values increasing by 0.021 ± 0.005 ‰ per km ($r = 0.46$, $p = 0.0001$)²⁹. On the Shatsky Rise, core-top foraminiferal $\delta^{18}\text{O}$ values from Site 1209 (this study) and 1208³⁰ show no significant differences in the last 25 ka ($p = 0.678$, Supplementary Fig. 3). Modern measurements of salinity and temperature on the Shatsky Rise show similar values at the two sites³¹ (Figure 2), with fresher (by 0.004 psu) and warmer (by 0.28°C) waters at Site 1209. From this, we conclude the Shatsky Rise was not bathed by a persistent NPDW in the recent past. (Lines 103 – 115)

In figure 2 the differences are associated with a shading that indicates the uncertainty. Although these shading are too faint to be properly appreciated, there is a clear superposition, especially in the estimated temperature of bottom waters. This type of superposition between the data clearly highlights that the differences are not so large. They are there, but they are not so clear. In my opinion this requires a little bit of statistical analyses to prove that the differences are statistically relevant (especially considering the variability of replicate analysis of each sample). On the other hand, from line 103 to 107 where the differences in $\delta^{18}\text{O}$ are discussed, the manuscript is quite generic:

“During MIS G4, G2, and the second half of MIS 104, there was a $<0.5^\circ\text{C}$ difference in BWT between the two sites, and a <0.1 ‰ difference in $\delta^{18}\text{O}$, compared to a >0.4 ‰ difference in $\delta^{18}\text{O}$ values in interglacials MIS G3, G1, and 103 (Figure 2)”

We would like to thank the reviewer for this comment and have addressed this in the manuscript, which we believe has made the paper better. As reflected in the changed title of the paper, we have changed the emphasis of the paper towards what the data

can show and reduced the weight given to speculation that can not be supported completely by the present data.

We have made the uncertainty envelopes on Figures 3 (previously Figure 2) and Figure 4 clearer so the reader can more easily see where the errors overlap.

We have also discussed, with use of statistics, where there are differences in BWT record as shown below, which replaced the previously more qualitative wording.

Our coupled Mg/Ca and $\delta^{18}\text{O}_{\text{benthic}}$ values show that the NPDW was colder than the underlying southern-sourced waters during the Late Pliocene (Figure 3). Our Mg/Ca derived bottom water temperature (BWT) values at Site 1209 are colder than BWT values at Site 1208³⁰ over the period 3.0 – 2.7 Ma, with a difference in the mean BWT of 0.45°C. (Lines 117 - 120)

When the BWT is coupled with the $\delta^{18}\text{O}_{\text{benthic}}$ values the colder Pliocene waters at Site 1209 must, therefore, have been fresher than those at the deeper Site 1208 to ensure density balance – in agreement with modelling evidence showing NPDW was fresher than underlying southern-sourced waters⁷ (Supplementary Fig. 4). However, the derived $\delta^{18}\text{O}_{\text{sw}}$ values at Site 1209 are significantly ($p < 10^{10}$, $N = 107$) heavier than those at Site 1208 by 0.24 ‰ in the Late Pliocene (Figure 3). (Lines 120 - 125)

The Mg/Ca record for the Early Pleistocene also suggests that there is a convergence in water mass properties in the North Pacific during glacials but not interglacials. Over the iNHG, the mean derived BWT at Sites 1209 and 1208 increased by $1.0 \pm 0.8^\circ\text{C}$ and $1.1 \pm 0.8^\circ\text{C}$ respectively (Figure 3). This is likely due to a greater influx of warm NADW into the Southern Ocean and a greater stratification of Southern Ocean surface waters following the iNHG which would warm the southern sourced waters flowing into the Pacific³⁰. Due to the reduction in NPDW formation after the iNHG, both sites would have experienced a greater share of Southern Ocean sourced waters which would have increased the BWT at both sites. While at Site 1209, there was effectively no difference in mean BWT between Early Pleistocene glacials and interglacials (0.7°C and 0.6°C respectively), Site 1208 saw a 0.5°C difference develop between mean glacial and interglacial BWT ($1.0 \pm 0.8^\circ\text{C}$ and $1.4 \pm 0.8^\circ\text{C}$ respectively). There was a slight decrease in the mean BWT gradient between Site 1209 and 1208 during Early Pleistocene glacials (0.3°C) and an increase in the mean BWT gradient during interglacials (0.8°C). This supports the idea of a convergence of water mass properties during glacials and a divergence during interglacials as seen in the $\delta^{18}\text{O}_{\text{benthic}}$ record, and lends credence to the idea that glacial-interglacial changes in the Early Pleistocene exerted a control on water mass formation in the Pacific that they did not during the Late Pliocene. (Lines 164 - 180)

Finally, we have discussed where in the record there is a quantitative difference in BWT record and where the current data can not completely support the current hypothesis.

Our results are constrained by the uncertainty in the BWT estimates, calculated by replicate analysis to be $\pm 0.8^{\circ}\text{C}$. This uncertainty is greater than the difference in Mg/Ca-derived BWT between 1209 and 1208, with only a few instances (e.g. MIS G9 and G8) during the Late Pliocene where the difference in the inferred temperatures is greater than the uncertainty (Figure 3). (Lines 128 - 132)

These results are caveated by the high uncertainties in BWT estimates ($\pm 0.8^{\circ}\text{C}$), however, there are stretches of MIS 101, 103, G1 and G3 when the BWT estimates at 1208 is warmer than 1209 outside of uncertainties (Figure 3) which is not seen during glacials, supporting the idea that there is some glacial-interglacial control on water mass properties in the deep North Pacific during the Pleistocene. (Lines 180 - 184)

As the whole paper is based on this evidence, I believe that a more extensive discussion and a more detailed statistical tests of the relevance of these differences is required. In this regard I think that Extended Data Figure 4 is much more effective in conveying the differences between the water masses. Extended Data Figure 4 includes only some data (maybe averages values? In the caption an explanation is not provided), however, a similar figure with all the data and their scatter, coupled with a statistical analysis would be much more effective in conveying the results of the paper. Indeed, following the brief discussion of the data from line 103 to line 107, the paper essentially becomes a review of literature data on various proxies related to ocean circulation during the Pliocene – Pleistocene interval. This discussion/review is much more extensive than the analysis of the data provided by the authors and relatively disconnected from the latter. Furthermore, this part is not properly organized and the narrative wanders between various subjects and elements (too many for a short paper) without a clear direction.

Thanks for this comment, we agree with the points here and have made adjustments accordingly. The paper has been updated to give a more quantitative basis to the claims made in the paper, as discussed above. We have also included a version of Extended Data Figure 4 in the paper as Figure 2, though with only some of the data from the study so as to maintain legibility. The figure caption on this figure has been updated to better convey this information to the reader.

Figure 2. Derived densities of past and modern water masses.

Temperature and salinity of past water masses from ODP site 1208 (green, 3346 m depth) and 1209 (orange, 2387 m depth) for Late Pliocene (stars), Early Pleistocene glacials (circles) and Early Pleistocene interglacials (triangles). The temperature is the mean of the BWT estimates for each site

over all intervals for the Late Pliocene (2.7 – 3.0 Ma) or the Early Pleistocene (2.4 – 2.7 Ma). The mean of all the $\delta^{18}\text{O}_{\text{sw}}$ estimates from the sites are used to determine the salinity estimates (see Methods). Isopycnals of σ_2 are calculated for 2300 m depth in the North Pacific using the Gibbs-Seawater Oceanographic Toolbox³² and shown in grey and labelled. The densities of waters at 1209 are greater than those at (deeper) site 1208 for the Late Pliocene and Early Pleistocene (hollow symbols) which suggests that the derived $\delta^{18}\text{O}_{\text{sw}}$ values for 1208 and/or 1209 cannot be linearly converted to past salinities. Modern values of temperature and salinity (diamonds) from Argo float data³¹, and calculated temperatures and salinities from core-top mean $\delta^{18}\text{O}_{\text{benthic}}$ and Mg/Ca values from 1209 and 1208³⁰ for the Holocene (0 – 12 ka, squares) are shown highlighting that modern and Holocene $\delta^{18}\text{O}_{\text{sw}}$ measurements do appear correlated to salinity (filled symbols).

The literature review on possible causes for water mass changes in the North Pacific has been generally cut down to a few lines in the introduction (see below), allowing for more of a focus on the data presented in the paper.

Active deep water formation in the North Pacific would require the removal of the halocline in the subpolar North Pacific, and studies have suggested a weak meridional (north-south) sea surface temperature (SST) gradient^{7,11}, changing monsoon patterns¹², and sea level induced changes in ocean gateways^{13, 14} as potential causes. (Lines 45 - 48)

In order to improve its impact the paper needs to be better organized: 1) Better and more clear presentation of the data 2) More detailed analysis of the data and their significance, using statistical methods to test the significance of the observed differences taking into account all the uncertainties. 3) Shorter and more impactful discussion of the existing literature to support the evidence obtained from the analysed dataset.

We would like to thank the reviewer again for their comments, which have been very helpful. We have changed the structure of the paper, firstly to reflect the Nature Communications style guide, but also in response to this review to better highlight the claims made by the evidence and to put less of an emphasis on the review of the previous literature.

We have included statistical tests on all the claims of difference between records for various intervals in this paper – some of which have highlighted that the changes are not as large as suggested in the original paper and the language of the paper has been adjusted to reflect this. In particular, the differences between the bottom water temperature records of the Sites 1209 and 1208 during the Early Pleistocene are shown to not be outside the analytical uncertainty and therefore cannot be said to show

unequivocally that there is a difference in water masses present at Site 1209 between glacial and interglacial in the Early Pleistocene. The presentation of this, and the conclusions from this about the causes of this water mass change over the iNHG, have been made more equivocal.

The literature review section of the paper has been moved to introduction and now focuses on those issues which are specifically pertinent to the data here presented, highlighting the challenges to the PMOC hypothesis as well as its adherents, to make clear how our data fits into the wider discussion around NPDW in the Pliocene.

There are other minor flaws that should also be addressed (how many samples were analysed? In certain figures, like extended figure 4, it is unclear if average values or selected values are being displayed). I have provided some other small notes in the annotated version of the manuscript.

The notes on other issues were welcomely received and the paper has been adjusted accordingly.

The small corrections on lines 14 – 18, line 20, and line 44 are no longer relevant as the relevant lines have been changed.

The keys for Figures 1, 3, and 4 are now also included in the figure as well as the in the figure caption.

The shading for Figure 2 (now Figure 3) has been made darker so as to be more visible.

The suggestion on Line 163 is no longer relevant as this section has been removed as it is no longer supported by the data.

The suggestions on Lines 178, 182, 198 and Lines 212 – 214 are no longer relevant as this section has been removed and is now included (in much shorter form) in the introduction.

The suggestions on Lines 226 – 227 are no longer relevant as this section of the paragraph has been removed to make the paragraph more focused.

Our coupled $\delta^{18}\text{O}_{\text{benthic}}$ and Mg/Ca values show that the NPDW was colder, and fresher than the underlying southern-sourced waters during the Late Pliocene (Figure 3). (Lines 117 – 118)

Schematic diagram showing ocean circulation in the Pacific Ocean during the Late Pliocene with mid Pliocene sea levels highlighting how an open Indonesian Gateway could promote NPDW formation. (Figure 6 Caption)

Stable Isotopes. 425 samples of the benthic foraminifera *Cibicidoides wuellerstorfi* was used for stable isotope analysis. (Lines 244 – 245)

The long term analytical reproducibility for NBS-19 is ± 0.06 ‰ and 0.05 ‰ (1 s.d.) for carbon and oxygen isotopes. (Lines 248 – 249)

135 samples of between 8 – 12 foraminifera of the species *Uvigerina peregrina* were picked for trace metal analysis. (Lines 256 - 257)

I hope that these suggestions might help the authors and the future impact of their paper to the general scientific community.

Reviewer #2 (Remarks to the Author):

Review of the manuscript “North Pacific Deep Water Formation controlled by Sea Level Changes”, by de Graaf et al.

The manuscript presents a very interesting benthic $\delta^{18}\text{O}$ record from the North Pacific (ODP 1209, at 2387 m depth) that, compared to the $\delta^{18}\text{O}$ record of a deeper site (ODP 1208, at 3345 m depth) from the same region, has been interpreted as a result of the presence of different water masses along the water column. The authors interpreted that ODP 1209 record indicates North Pacific deep water formation was active before the intensification of the Northern Hemisphere glaciation. They also linked the cessation of North Pacific deep water formation to the restriction in the water mass transport through the Indonesian Gateway, as suggested by ocean modelling studies. The hypothesis proposed is interesting, however, I do not clearly see in the presented graphics a dataset that supports a clear cessation of North Pacific deep water formation after the intensification of the Northern Hemisphere glaciation. Figure 1 shows the difference benthic $\delta^{18}\text{O}$ difference between the two sites as a proxy for the strength of North Pacific deep water formation but there is no marked reduction.

We would like to thank the reviewer for their helpful comments. As the new proposed title of the paper shows, we have changed the paper to focus on the evidence for NPDW formation in the Late Pliocene as well as the cessation in the Early Pleistocene, and reduced the focus on mechanisms for the changes in NPDW formation which are less supported by the data of the paper, and we believe that the paper is now more data-driven than previously.

We have changed Figure 1, to better show both the deep water formation prior to the INHG and the reduction in NPDW strength during Early Pleistocene glacials. We have simplified Figure 1 to better highlight how the glacial $\Delta\delta^{18}\text{O}$ is smaller than the interglacial $\Delta\delta^{18}\text{O}$, seen in the blue 10-kyr smoothed average of $\Delta\delta^{18}\text{O}$, and have

removed the lines connecting the binned average $\Delta\delta^{18}\text{O}$ values as this complicated rather than clarified the figure.

We have softened the language around the change in the $\delta^{18}\text{O}_{\text{benthic}}$ before and after the iNHG. Our new statistics analyses do show there is a decrease in the gradient, but not a complete collapse in the gradient between the two sites during the glacials.

During the Late Pliocene (3.3 – 2.7 Ma), the $\delta^{18}\text{O}$ values at the two sites show a mean difference of 0.25‰, but this is regularly in excess of 0.4 ‰ (Figure 1). This is a significant difference in the $\delta^{18}\text{O}$ values ($p < 10^{-31}$, $N = 601$) far in excess of the measurement uncertainty of ± 0.05 ‰ and does not vary significantly between glacials and interglacials over the Late Pliocene ($p = 0.8110$, $N = 174$). (Lines 91 – 95)

After iNHG (c. 2.7 Ma), the $\delta^{18}\text{O}_{\text{benthic}}$ records on the Shatsky Rise continue to be offset during interglacials (Figure 1) and converge during glacials (Figure 3) indicating that NPDW formation may be suppressed during post-iNHG glacials. From MIS G6 onwards, the mean $\delta^{18}\text{O}_{\text{benthic}}$ difference decreased to 0.15 ‰ during Early Pleistocene glacials from 0.25 ‰ in the Late Pliocene (Figures 1, 5). During interglacials, however, the mean difference in $\delta^{18}\text{O}_{\text{benthic}}$ values actually increases to 0.32 ‰ during the Early Pleistocene, an increase in 0.07 ‰ from the Late Pliocene (Figures 1, 5). The difference in mean $\delta^{18}\text{O}_{\text{benthic}}$ shows no significant difference (< 0.01 ‰, $p = 0.811$) between glacials and interglacials in the Late Pliocene, but a significant difference (0.17 ‰, $p = 0.002$) between glacials and interglacials in the Early Pleistocene. (Lines 147 – 155)

We have also included Figure 5, which shows the distribution of $\Delta\delta^{18}\text{O}$ values in both glacials and interglacials and highlights how the glacial $\Delta\delta^{18}\text{O}$ is considerably reduced compared to the interglacial difference.

The discussion about BWT is not easy to follow in figure 2, a gradient would be better to show if there is a difference between both sites that changed at the intensification of the Northern Hemisphere glaciation. Same thing for $\delta^{18}\text{O}_{\text{sw}}$.

This is a good point, and we have changed the discussion of the BWT and $\delta^{18}\text{O}_{\text{sw}}$ to make it clearer what points are being made. We have included some statistical tests to highlight the size and significance of any of the changes. We have focussed the discussion on the qualitative differences (such as in the mean BWT) that can be seen between the two records which should make the discussion easier to follow. Additionally, we have included a gradient in the BWT and $\delta^{18}\text{O}_{\text{sw}}$ in the supplementary information as Supplementary Figure 7.

When the BWT is coupled with the $\delta^{18}\text{O}_{\text{benthic}}$ values the colder Pliocene waters at Site 1209 must, therefore, have been fresher than those at the deeper Site 1208 to ensure density balance – in agreement with modelling evidence showing NPDW was fresher than underlying southern-sourced waters⁷ (Supplementary Fig. 4). However, the derived $\delta^{18}\text{O}_{\text{sw}}$ values at Site 1209 are significantly ($p < 10^{-10}$, $N = 107$) heavier than those at Site 1208 by 0.24 ‰ in the Late Pliocene (Figure 3). (Lines 120 - 125)

The Mg/Ca record for the Early Pleistocene also suggests that there is a convergence in water mass properties in the North Pacific during glacials but not interglacials. Over the iNHG, the mean derived BWT at Sites 1209 and 1208 increased by $1.0 \pm 0.8^\circ\text{C}$ and $1.1 \pm 0.8^\circ\text{C}$ respectively (Figure 3). This is likely due to a greater influx of warm NADW into the Southern Ocean and a greater stratification of Southern Ocean surface waters following the iNHG which would warm the southern sourced waters flowing into the Pacific³⁰. Due to the reduction in NPDW formation after the iNHG, both sites would have experienced a greater share of Southern Ocean sourced waters which would have increased the BWT at both sites. While at Site 1209, there was effectively no difference in mean BWT between Early Pleistocene glacials and interglacials (0.7°C and 0.6°C respectively), Site 1208 saw a 0.5°C difference develop between mean glacial and interglacial BWT ($1.0 \pm 0.8^\circ\text{C}$ and $1.4 \pm 0.8^\circ\text{C}$ respectively). There was a slight decrease in the mean BWT gradient between Site 1209 and 1208 during Early Pleistocene glacials (0.3°C) and an increase in the mean BWT gradient during interglacials (0.8°C) supporting the idea of a convergence of water mass properties during glacials and a divergence during interglacials as seen in the $\delta^{18}\text{O}_{\text{benthic}}$ record, lending credence to the idea that glacial-interglacial changes in the Early Pleistocene exerted a control on water mass formation in the Pacific that they did not during the Late Pliocene. (Lines 164 - 180)

We have also included a section discussing the limitations of the differences in the BWT estimates and what can, and more importantly, what cannot be inferred from our record.

Our results are constrained by the uncertainty in the BWT estimates, calculated by replicate analysis to be $\pm 0.8^\circ\text{C}$. This uncertainty is greater than the difference in Mg/Ca-derived BWT between 1209 and 1208, with only a few instances (e.g. MIS G9 and G8) during the Late Pliocene where the difference in the inferred temperatures is greater than the uncertainty (Figure 3). (Lines 128 - 133)

These results are caveated by the high uncertainties in BWT estimates ($\pm 0.8^\circ\text{C}$), however, there are stretches of MIS 101, 103, G1 and G3 when the BWT estimates at 1208 is warmer than 1209 outside of uncertainties (Figure 3) which is not seen during glacials, supporting the idea that there is some glacial-interglacial control on water mass properties in the deep North Pacific during the Pleistocene. (Lines 180 - 184)

We have included a gradient in the BWT and $\delta^{18}\text{O}_{\text{sw}}$ in the supplementary information as Supplementary Figure 7. This is not included in the main paper as it was deemed to make the Figure 3 too confusing, particularly as, due to poor temporal overlap between the records, any attempt to determine a gradient had a much lower resolution than the original records.

Figure 2d is intriguing, it seems that instead of a cessation in North Pacific deep water formation, this process was enhanced during interglacials after 2.7 Ma. The support for the presence/absence of North Pacific deep water formation must be clearly explained indicating the differences between the Southern Ocean deep water and the NPDW.

This is a really good point. We have used statistical tests to show the magnitude of the difference in BWT between glacial and interglacials. However, as this difference is within the analytical uncertainty we have also changed the language and wording around the cessation of NPDW after the iNHG to be more equivocal.

a greater influx of warm NADW into the Southern Ocean and a greater stratification of Southern Ocean surface waters following the iNHG which would warm the southern sourced waters flowing into the Pacific^{30, 35}. Due to the reduction in NPDW formation after the iNHG, both sites would have experienced a greater share of Southern Ocean sourced waters which would have increased the BWT at both sites. While at Site 1209, there was effectively no difference in mean BWT between Early Pleistocene glacial and interglacials (0.69°C and 0.63°C respectively), Site 1208 saw a 0.5°C difference develop between mean glacial and interglacial BWT (0.99°C and 1.4°C respectively). There was a slight decrease in the mean BWT gradient between Site 1209 and 1208 during Early Pleistocene glacial (0.30°C) and an increase in the mean BWT gradient during interglacials (0.79°C) supporting the idea of a convergence of water mass properties during glacial and a divergence during interglacials as seen in the $\delta^{18}\text{O}_{\text{benthic}}$ record, lending credence to the idea that glacial-interglacial changes in the Early Pleistocene exerted a control on water mass formation in the Pacific that they did not during the Late Pliocene. These results are caveated by the high uncertainties in BWT estimates ($\pm 0.8^\circ\text{C}$), however, there are stretches of MIS 101, 103, G1 and G3 when the BWT estimates at 1208 is warmer than 1209 outside of uncertainties (Figure 3) which is not seen during glacial, supporting the idea that there is some glacial-interglacial control on water mass properties in the deep North Pacific during the Pleistocene. (Lines 167 – 184)

We highlight the greater interglacial difference in $\delta^{18}\text{O}$ after the iNHG in Figure 5 as well. However, we do not expand on this idea too much as we thought that to fully discuss this idea would complicate the paper too much and distract from the main findings of the paper. We agree that it is interesting that there is an increase in the interglacial $\Delta\delta^{18}\text{O}$ as well as a decrease in the glacial $\Delta\delta^{18}\text{O}$ but found that it was too difficult to tie in with the rest of the paper without stretching the paper too thin.

Figure 2d was also removed from Figure 2 (now Figure 3 in the updated manuscript) as it was thought to complicate the figure too much, and the glacial/interglacial mean $\Delta\delta^{18}\text{O}$ was calculated from the binned data from Figure 1 which introduced uncertainty into the figure which was not communicated effectively.

In addition, the text tries to justify the hypothesis from the very beginning. I think it would be better to present first the modern conditions and water masses characteristics; then indicate how the interval before the intensification of the Northern Hemisphere glaciation shows different conditions. The discussion about possible

drivers for this change is also repeated in two different sections. I suggest reading again all the manuscript and try to arrange the story and the data in a clearer way.

Many thanks for this comment, we have taken this on board and have restructured the paper accordingly and in line with Nature Communications style guide. Firstly, the introduction has been rewritten to be more even-handed on the concept of a Pliocene PMOC.

The evidence for NPDW formation in the Late Pliocene is contested and relies heavily on proxy records that are easily biased by changes in local productivity. Many modelling studies of the Late Pliocene fail to generate an active overturning circulation in the North Pacific¹⁵; though this may be due to limited model run time⁶, poor reconstructions of Late Pliocene temperature gradients⁷, or an inability of the models to accurately simulate changing Pliocene palaeogeography¹⁴. Modelling studies that alter atmospheric cloud physics to simulate a weak meridional SST gradient suggested by proxy data during the Pliocene generate active NPDW formation in the Pliocene⁷. This water mass structure is supported by the arrangement of carbon isotopes across the Late Pliocene Pacific Ocean which point to NPDW reaching depths of around 3000 m in the North Pacific⁶, though recent work from the subpolar North Pacific questions this¹⁶. Critically, these records come from opposite sides of the Pacific Ocean and NPDW formation is expected to occur in the Northwest Pacific Ocean during the Pliocene and is unlikely to show a similar water mass structure in the Northeast Pacific^{9,17}. Carbon isotopes are also readily influenced by changes in local productivity¹⁸ and so it is hard to determine which, if any, of these signals are driven by water mass changes. The presence of NPDW in the Pliocene is also supported by opal and carbonate mass accumulation rates^{19,20} and redox-sensitive trace metal records⁷ in the subpolar North Pacific, which both suggest an unstratified and well-ventilated water column indicative of active deep water formation as well as upwelling of deep waters in the Late Pliocene⁷. (Lines 50 – 67)

Secondly, we have reduced the discussion on the possible drivers for NPDW formation to instead focus on the data that is presented here in the paper. The possible theories are discussed briefly in the introduction:

Active deep water formation in the North Pacific would require the removal of the halocline in the subpolar North Pacific, and studies have suggested a weak meridional (north-south) sea surface temperature (SST) gradient^{7,11}, changing monsoon patterns¹², and sea level induced changes in ocean gateways^{13,14} as potential causes. (Lines 45 – 48)

And then the discussion at the end of the paper is much reduced (Lines 194 – 211) and focusses on what can be conclusively drawn from the data and removes much of the discussion of alternatives as seen in the original paper (Lines 169 – 202 in the original paper, now removed).

The following minor comments have been addressed:

Minor comments

Line 57: I believe the authors wanted to report that “A positive difference indicates more positive $\delta^{18}\text{O}$ values at Site 1209” (not in 1208)

A positive difference indicates more positive $\delta^{18}\text{O}$ values at Site 1209. (Figure 1 Caption)

Line 63: specify benthic foraminifers

This has been fixed and greater clarity added throughout the paper that the stable isotope measurements have been taken from benthic foraminifera rather than simply foraminifera.

Note: Reviewer comments are in green text, author replies are in black text, and manuscript edits are in black text with track changes. The annotated line numbers refer to those lines in the final document.

We would like to thank the reviewers for their helpful feedback, which we have taken on board, and which have improved the paper. We have changed wording where this is ambiguous and corrected any typos highlighted by the reviewers. In response to the third reviewer, we have included a new figure (Figure 1) which provides useful context on the location of the sites within the North Pacific and in relation to the modern distribution of water masses in the North Pacific. We have also clarified the wording of the abstract to better highlight the main points of the article.

A point-by-point response to the reviewer comments is included below.

Reviewer #1 (Remarks to the Author):

The revised version of “Reduced North Pacific Deep Water Formation across the Northern Hemisphere Glaciation” significantly improved the shortcomings of the original version. The data are now better discussed, associated with a statistical analysis, and supported by clear figures, making the message conveyed by the manuscript much clearer and more impactful.

I believe that the paper is now suitable for publication. I have noted some small typos here and there in the annotated version of the manuscript.

I hope that this small correction can help the authors, and I hope to see the manuscript published soon.

Best regards

We would like to thank the reviewer for their comments and their help in improving this manuscript. We have corrected the typos as below:

Evidence suggests there was a Pacific Meridional Overturning Circulation during the warm Late Pliocene, similar to the modern Atlantic Ocean with a weak halocline in the subpolar North Pacific resulting in North Pacific Deep Water (NPDW) formation. (Lines 15 – 18)

Modelling studies that alter cloud physics to simulate a weak meridional Pacific SST gradient, as proxy suggest was the case during the Pliocene¹¹, generate active NPDW formation. (Lines 56 – 57)

Therefore, $\delta^{18}\text{O}$ in benthic foraminifera ($\delta^{18}\text{O}_{\text{benthic}}$) can be used as a proxy for water mass density (Lines 74 – 75)

foraminiferal $\delta^{18}\text{O}$ values from Site 1209 (this study) and 1208³⁰ show no significant differences during the last 25 ka (Lines 112 – 113)

in agreement with modelling evidence showing that the NPDW was fresher (Lines 123 – 124)

However, our BWT record shows that the temperatures measured at Site 1209 were consistently colder (Lines 138 – 139)

indicating that NPDW formation may have been suppressed during post-iNHG glacials (Lines 147 – 148)

During interglacials, however, the mean difference in $\delta^{18}\text{O}_{\text{benthic}}$ values actually increases to 0.32 ‰ during the Early Pleistocene, an increase of 0.07 ‰ from the Late Pliocene (Lines 150 – 151)

The difference in mean $\delta^{18}\text{O}_{\text{benthic}}$ shows no significant difference (< 0.01 ‰, $p = 0.811$, $N = 174$) between glacials and interglacials in the Late Pliocene, but a significant difference (0.17 ‰, $p = 0.002$, $N = 107$) between glacials and interglacials during the Early Pleistocene. (Lines 152 – 154)

however, there are stretches of MIS 101, 103, G1 and G3 where the BWT estimates at 1208 are warmer than 1209 outside of uncertainties (Lines 179 – 180)

We suggest falling sea levels in Early Pleistocene glacials would restrict water mass transport through the Indonesian Gateway and act as a control on the formation of NPDW (Lines 192 – 193)

Similarly, planktic oxygen isotope records from the subpolar North Pacific imply local salinity changes, proportional to the strength of the subpolar halocline, in phase with global sea level changes after 2.7 Ma⁴³. The Indonesian Gateway thus links changes in sea level to the North Pacific meridional SST gradient and subpolar surface salinity (Lines 205 – 208)

to give the correction added to the salinity estimate to achieve the final salinity estimate. (Lines 293 – 294)

Schematic diagram showing ocean circulation in the Pacific Ocean during glacials of the Early Pleistocene with sea levels 60 m lower than modern (Lines 542 – 543)

Elevation above 1000 m is shown in brown. Location of ODP Sites 1209 (orange) and 1208 (blue) are shown as circles. (Lines 552 – 553)

Reviewer #3 (Remarks to the Author):

Response:

In regard to the North Pacific Deep Water formation in the Late Pliocene, there is some controversial on the proxy and model climate. In this work, Dr. Graaf et al. present a coupled Mg/Ca- $\delta^{18}\text{O}$ record from the North Pacific which shows two distinct water masses in the Pliocene North Pacific Ocean, with NPDW colder and fresher than the

underlying deeper water. Using two cores from different water depth, they further prove the NPDW formation decline during glacials from 2.7 million years ago, which highlight the impact of sea level via ocean gateways in shaping deep water circulation in the climate system. After the new version and response letter, I think the authors have gave a good reply and made the modifications in the manuscript. I give some comments on this work.

Firstly, in the Abstract part: I suggest that the author should add some highlights in the last sentence in this part. In this version, I cannot get your key points in your study. Secondly, I suggest that you should add a figure 1 about the location in the North Pacific Ocean. In this case, you can present the deep-water circulation in the North Pacific Ocean.

After some improvement in this part, I suggest this manuscript can be accepted after the minor revision.

We would like to thank the reviewer for their comments. We have reworded the final line of the abstract as follows:

Here, we show a decline in NPDW formation during glacials from 2.7 million years ago, which we suggest demonstrates the strong sensitivity of ocean gateways to sea level and ice volume in shaping deep water circulation, and thus the wider climate system. (Lines 21 – 24).

We have also included a new figure, Figure 1, which shows both the location of Sites 1208 and 1209 in the North Pacific and the paths of major water masses in the deep North Pacific. This should make it clearer to readers where the sites are located and how they relate to modern ocean circulation.

Figure 1. Location of ODP Sites 1208 and 1209 and major water masses in the modern Pacific Ocean.

Salinity of the modern Pacific Ocean below 1000 m depth along WOCE Transect P13 running at 165°E from 4°S to 54°N⁶¹. Locations of the Ocean Drilling Program (ODP) Sites 1208 and 1209 are shown as circles. Major water mass trajectories in the North Pacific are shown with white arrows. AABW = Antarctic Bottom Water, CDW = Circumpolar Deep Water, NPIW = North Pacific Intermediate Water. Bathymetry derived from eWOCE online archive⁶². (Lines 482 – 488)